# Mucopolysaccharidosis Type I: Current Treatments, Limitations, and Prospects for Improvement

**DOI:** 10.3390/biom11020189

**Published:** 2021-01-29

**Authors:** Christiane S. Hampe, Jacob Wesley, Troy C. Lund, Paul J. Orchard, Lynda E. Polgreen, Julie B. Eisengart, Linda K. McLoon, Sebahattin Cureoglu, Patricia Schachern, R. Scott McIvor

**Affiliations:** 1Immusoft Corp., Seattle, WA 98103, USA; jake.wesley@immusoft.com; 2Department of Pediatrics, University of Minnesota, Minneapolis, MN 55455, USA; lundx072@umn.edu (T.C.L.); orcha001@umn.edu (P.J.O.); eisen139@umn.edu (J.B.E.); 3The Lundquist Institute at Harbor, UCLA Medical Center, Torrance, CA 90502, USA; lpolgreen@lundquist.org; 4Department of Ophthalmology and Visual Neurosciences, University of Minnesota, Minneapolis, MN 55455, USA; mcloo001@umn.edu; 5Department of Otolaryngology, Head and Neck Surgery, University of Minnesota, Minneapolis, MN 55455, USA; cureo003@umn.edu (S.C.); schac002@umn.edu (P.S.); 6Immusoft Corp, Minneapolis, MN 55413, USA; mcivo001@umn.edu; 7Department of Genetics, Cell Biology and Development and Center for Genome Engineering, University of Minnesota, Minneapolis, MN 55455, USA

**Keywords:** mucopolysaccharidosis type I, Hurler syndrome, enzyme replacement therapy, hematopoietic stem cell transplantations, animal models, experimental therapies

## Abstract

Mucopolysaccharidosis type I (MPS I) is a lysosomal disease, caused by a deficiency of the enzyme alpha-L-iduronidase (IDUA). IDUA catalyzes the degradation of the glycosaminoglycans dermatan and heparan sulfate (DS and HS, respectively). Lack of the enzyme leads to pathologic accumulation of undegraded HS and DS with subsequent disease manifestations in multiple organs. The disease can be divided into severe (Hurler syndrome) and attenuated (Hurler-Scheie, Scheie) forms. Currently approved treatments consist of enzyme replacement therapy (ERT) and/or hematopoietic stem cell transplantation (HSCT). Patients with attenuated disease are often treated with ERT alone, while the recommended therapy for patients with Hurler syndrome consists of HSCT. While these treatments significantly improve disease manifestations and prolong life, a considerable burden of disease remains. Notably, treatment can partially prevent, but not significantly improve, clinical manifestations, necessitating early diagnosis of disease and commencement of treatment. This review discusses these standard therapies and their impact on common disease manifestations in patients with MPS I. Where relevant, results of animal models of MPS I will be included. Finally, we highlight alternative and emerging treatments for the most common disease manifestations.

## 1. Introduction

Mucopolysaccharidosis type I (MPS I) is a lysosomal disease, caused by a deficiency of the enzyme alpha-L-iduronidase (IDUA). Currently approved treatments consist of enzyme replacement therapy (ERT) and/or hematopoietic stem cell transplantation (HSCT). While these treatments significantly improve disease manifestations and prolong life, a considerable burden of disease remains. Both treatments may at best prevent the development or worsening of abnormal function and somatic complications but cannot revert already existing symptoms. Therefore, treatment must commence as early as possible for maximum effect and diagnostic delay-due to the nonspecific nature of early symptoms-limits treatment success [1]. To overcome this limitation, implementation of MPS I in newborn screening programs is strongly recommended [2,3].

Moreover, the therapeutic effect in some systems appears to wear off after several years. Bones, eyes, and heart valves prove to be especially resistant to treatment. A number of experimental strategies are currently under development to reduce the remaining burden of disease.

## 2. Standard Therapies for MPS I

### 2.1. Allogeneic HSCT

Allogeneic HSCT is considered the gold standard for treatment of Hurler syndrome and can alleviate a number of disease symptoms and increase the patient’s life span, especially when performed before the age of 2 years and prior to cognitive impairment [4,5,6,7,8]. Clinical effect is evident in reduced facial coarseness, joint mobility, and reduction in sleep apnea, cardiac disease, and hearing loss [9,10,11]. Overall, survival is significantly prolonged [4,12,13], and when initiated early, decline of neurocognition can be stabilized [8]. However, clinical benefit varies, both between patients and between different organs in the same patient. Interpatient variation is mainly due to differential effectiveness of engraftment and is governed by such variables as genotype, age at transplantation, and donor specifications [14]. Tissue specific improvement is caused by different accessibility of tissues, such as heart, eyes, and bones, to circulating enzyme. Notably, HSCT can prevent, but not significantly improve, clinical manifestations in bone, cornea, cardiac valves, and CNS [15,16], resulting in residual disease burden in the majority of patients.

#### 2.1.1. Mortality Rates and Conditioning Regimens

Improved myeloablative conditioning regimens, donor matching availability, and improved supportive care have greatly reduced mortality rates associated with HSCT for MPS I during the last decades at experienced centers [5]. Viral infections, graft rejection, pulmonary hemorrhage, and graft-versus-host disease (GvHD) remain the most common causes of death and are most commonly seen in the first several months after transplant, but can occur within the first year post-transplantation [13,17]. These complications may be related to the immunologic responses of the recipient to the donor cells or vice versa, or may be due to the conditioning chemotherapy regimen, with use of agents such as busulfan and cyclophosphamide [5,18]. Conditioning is necessary for successful engraftment, allowing complete donor chimerism and creation of niches for the incoming donor stem cells. Myeloablative conditioning and reduced-intensity conditioning regimens are available. In the early era of HSCT, patients were subjected to total body irradiation (TBI) with severe side-effects, including neurocognitive impairment, stunted growth, and a higher risk of hypothyroidism and cataracts, especially in very young children [19,20]. Long-term effects of myeloablative conditioning regimens are infertility in both females and males [21,22]. Comparing infertility risks associated with treatment in pre-puberty (1–12 years) with treatment after the age of 13 revealed opposite trends in males and females. While pre-pubertal treatment in males was associated with increased risk for infertility, females treated at a younger age had a lower risk for infertility [23]. Busulfan-based treatment was associated with higher infertility in females but not in males, and TBI increased risk of infertility in males, but not in females [23]. Fertility preservation needs to be addressed in patients prior to HSCT [24].

Recent transplant regimens utilize non-TBI based preparative conditioning regimens. Current regimens are generally based on a combination of busulfan and fludarabine or cyclophosphamide [5,25,26]. While the use of busulfan is an important agent to achieve successful transplants with high chimerism, the myeloablative drug is associated with significant toxicity [20,27]. Reduced intensity conditioning regimens may decrease toxicity, but are associated with an increased risk for graft failure [17]. Antibody based therapy, such as anti-thymocyte globulin (ATG) or anti-CD52 antibody therapy are being utilized as a means of decreasing toxicity in combination with reduced intensity regimens [28], and continuing research is ongoing to optimize an approach that achieves full donor chimerism while minimizing side-effects.

#### 2.1.2. Effectiveness of HSCT

Achieving the best outcomes in patients treated by HSCT depends on a variety of factors, including the age of the recipient, existing disease manifestations, the availability of a well matched donor, donor status (carrier or non-affected), and tissue source of HSCT [29]. Different biochemical parameters can be used to evaluate success of HSCT, such as glycosaminoglycan (GAG) levels in blood, spinal fluid, and urine, IDUA activity levels in blood [30,31], and leukocyte lysates [32]. Umbilical cord blood has largely replaced BM as donor cell sources in young patients with MPS I. Umbilical cord blood presents an attractive alternative because of the less stringent requirements for donor HLA-matching and ease of availability [7,33].

The level of enzyme activity directly correlates with donor chimerism and the use of a non-carrier donor [5,31,33]. In patients with full donor chimerism normal IDUA activity can be measured in blood cell lysates, but delivery of enzyme to the tissues is more difficult to ascertain [34]. Rapid reduction of heparan sulfate and dermatan sulfate in blood and urine is observed in the majority of patients [1,11,31,35,36,37]. This reduction is long-lived and continues for years after transplantation. However, GAG levels typically remain above reference levels [36], although normal leukocyte lysate enzyme levels are observed for the majority of patients [1,36,37]. The lack of normalization of blood and urine levels may be a consequence of partial GAG degradation in difficult-to-reach organs. Remaining GAGs will diffuse from these organs into the circulation and will eventually be excreted in the urine [36]. In support of this, dermatan sulfate stems predominantly from hard-to-reach organs and tissues, such as bone, cartilage, and heart valves [38,39] and dermatan sulfate levels tend to remain relatively elevated compared to heparan sulfate [36].

Animal studies further confirmed that IDUA expression and GAG level reduction differ between tissues. HSCT performed in young MPS I cats resulted in significant increases in IDUA activity in liver, spleen, lung, and thyroid, but not in kidney, brain, or heart. GAG levels were reduced in liver, lung, thyroid, kidney, and heart. The reduction of GAG levels in the absence of detectable IDUA activity in kidney and heart may indicate that IDUA levels below the sensitivity of the activity assay were sufficient to reduce GAG levels [40]. In neonatal MPS I mice HSCT resulted in significant increases of IDUA activity, most prominently in the spleen, followed by liver and kidneys, heart, and lungs. While IDUA activity also varies between tissues in wild-type animals, IDUA levels after HSCT reached up to 70% of normal in the spleen but only 10% of normal in the lungs and heart. Consequently, reduction in GAG levels were also found to be tissue dependent. While GAG levels in spleen and liver were normalized overall, they were only partially reduced in kidneys, heart, and lungs [41,42].

This tissue specificity is probably a consequence of differential donor cell engraftment and differentiation to resident cells, and diffusion or lack thereof of circulating enzyme. Donor monocytes can leave the blood stream and infiltrate into peripheral organs, where they differentiate into tissue-specific macrophages [29]. Enzyme produced by engrafted donor cells can be transferred to the recipient’s cells via cross-correction. Enzyme is released by donor macrophages and leukocytes and is taken up by the recipient’s cells, where GAGs are metabolized [43,44].

Thus, organs with higher vascularization, such as liver, and spleen, show higher numbers of engrafted donor cells, while tissues with less vascularization, including cartilage and corneas, will benefit less from the transplanted cells. The brain poses a unique situation, where donor monocytes differentiate into microglia or brain macrophages after crossing the blood brain barrier (BBB) [45]. Replacement of the recipients’ microglia by donor-derived cells takes up to one year [46,47]. During the first posttransplant year, deterioration in intellect and development may continue, emphasizing the importance of early treatment to prevent cognitive impairment [11,48,49,50].

### 2.2. Enzyme Replacement Therapy (ERT)

ALDURAZYME (laronidase) is a recombinant variant of the human IDUA produced in a Chinese hamster ovary cell line. The ~82 kD glycoprotein contains mannose-6-phosphate (M6P) residues that enable the binding of the enzyme by cell surface M6P receptors. Subsequent uptake directs the enzyme to the lysosomes. Intravenous infusions with laronidase are administered weekly. Complications are rare, and ERT is considered safe overall [51]. Weekly intravenous infusions of 0.58 mg/kg of laronidase led to a significant drop in urinary GAG levels [51,52,53,54] accompanied by a reduction in hepatosplenomegaly [51,52,54,55]. ERT also improved upper airway restrictions, physical performance, and resulted in some improvement in left ventricular hypertrophy [56,57,58].

While ERT for patients with MPS I is well tolerated with no serious adverse events, the infusions require several hours every week, adding to the disease burden of patients and families. Another major drawback lays in the enzyme’s low level of BBB penetration and inefficient delivery to avascular tissues [59,60]. Consequently, cognitive function, skeletal deformities, and visual acuity do not improve when it is the sole therapy [61,62,63,64,65,66]. Moreover, the majority of patients (up to 90%) produce IgG antibodies to laronidase in response to ERT [65], which can interfere with enzyme activity and uptake [65,67]. Studies investigating the effect of ERT-induced IDUA antibodies on biochemical markers and clinical parameters gave conflicting results [56]. While all studies concluded that ERT led to the development of IDUA antibodies and poorer urinary GAG reduction [30,51,65,67], one study found no effect on clinical outcome in patients receiving only ERT [65]. Another study found a higher incidence of sleep disordered breathing in ERT-treated patients with or without HSCT, probably caused by IDUA antibody development [32]. Interestingly, these antibodies diminish over time despite continuous ERT [51,65,68].

Contradictory results were reported when development of antibodies to IDUA were examined. In some cases post-transplant ERT was associated with the development of IDUA antibodies, which were inhibitory and were accompanied with an increase in heparan sulfate excretion [67,69] and poorer endurance as established by the 6-min-walk-test [67]. However, in the absence of inhibitory antibodies, ERT post-HSCT had a beneficial effect on the 6-min walk test [67]. Therefore, efforts are currently focused on the induction of tolerance to exogenous IDUA [70,71,72].

While ERT is not recommended as the sole treatment for Hurler syndrome, a combination of ERT with HSCT may have benefits over each treatment alone [73,74,75]. Peri-transplant ERT appears to have beneficial effects on the clinical condition of the patient [7,31,48,50,58,67,73,74,76]. ERT can bridge the time until a suitable donor for HSCT has been identified, and therefore ERT is often initiated at time of diagnosis [31]. Importantly, ERT was not associated with a reduced engraftment rate [48,73,76,77], and subsequent HSCT attenuated the formation of neutralizing IDUA antibodies [78]. Moreover, GAG-reduction due to peri-transplant ERT appeared to improve HSCT engraftment [73,75,79]. Continuous ERT post-transplant has been reported to improve residual disease burden [48,56,65,72,74]. Recent studies in neonatal MPS I mice allowed an in-depth analysis of the combined treatment [80]. Animals receiving both HSCT and ERT showed higher IDUA levels in the spleen, lower plasma GAG levels, and improved bone architecture compared to animals receiving either treatment alone [80]. Beneficial effects of combined HSCT/ERT treatment pertaining to specific manifestations are discussed in the appropriate sections below.

## 3. Impact of HSCT and ERT on Tissue-Specific Disease Manifestations

The effect of HSCT, ERT, and the combination of both treatments on different disease manifestations is summarized in Table 1.

### 3.1. Ocular Manifestations

Ocular manifestations of MPS I include corneal clouding, retinal degeneration, optic nerve damage, and glaucoma. The effects of HSCT on ocular manifestations are complex. The cornea is an avascular structure, and in general not protected by HSCT. A multinational study of over 200 patients with MPS I, with a follow-up period of over 9 years, showed stabilization or improvement of corneal clouding for the majority of patients [5]. While several earlier studies supported this finding [81,82,83], others showed onset of corneal clouding despite HSCT [64,84,85]. A recent longitudinal analysis of 24 patients with MPS I indicated stabilization of corneal clouding during the first years post HSCT, after which corneal clouding reappeared [86]. The differences in outcomes may be due to sample size, length of follow-up, success of engraftment, and post-transplant enzyme levels. Both Aldenhoven et al. [5] and Javed et al. [84] noticed a strong correlation between outcome, age at transplant, level of engraftment, and post-transplant IDUA levels. Recent studies detected large numbers of myofibroblasts in the cornea of a MPS I patient after HSCT, indicating that continuous or reappearing of corneal clouding may be caused by the transformation of keratocytes into myofibroblasts, which would not be affected by HSCT [87].

While initial analyses of ocular symptoms other than corneal clouding showed promising effects [82], the majority of later studies reported continued loss of visual acuity and increased retinal dysfunction despite HSCT [11,35,81,85,86,88,89]. As the retina is part of the central nervous system, it is protected from blood cell entry by the blood-retinal barrier. Thus, over time after HSCT treatment, visual acuity declined and retinal pathology visible on OCT examination developed [64].

Animal studies revealed species-specific responses in the eye to both HSCT and ERT. In dogs, HSCT resulted in greatly reduced development of ocular manifestations, including corneal clouding [90,91]. Dogs also demonstrated a beneficial effect of ERT on GAG accumulation in the cornea [92], while no significant effect on ocular manifestations in patients with MPS I or mice was noted in response to ERT [93,94]. Results from ERT in MPS I cats were largely inconclusive, as corneal clouding was reduced in only one of two cats treated with high-dose ERT [54].

One possible reason for the lack of ocular responsiveness to ERT may be due to insufficient delivery of enzyme to the eye. Low levels of IDUA in tear film indicate poor enzyme transport from the circulation to multiple tissues within the orbit [95].

It is important to note that there is a great deal of variability in the ocular pathology in individuals with MPS I [96]. Due to the inability of HSCT and ERT to prevent retinal and corneal pathology, it is critical to detect and treat patients with MPS as early as possible in order to improve their eyesight long-term [97].

### 3.2. Respiratory System

Upper airway obstructions often result in sleep disordered breathing (SDB), including obstructive sleep apnea syndrome (OSAS). HSCT appeared to improve SDB, including OSAS, through reduction in adenoid hyperplasia, tongue, and maxillary constrictions [32,98,99,100,101]. However, a long-term follow-up study of 10 years indicated that the beneficial effect of HSCT may be only temporary [102]. Similar to other disease manifestations, effectiveness of HSCT on respiratory function depended on non-carrier donor and enzyme levels one year post HSCT [32]. Improvement of respiratory symptoms, even if only temporary, benefited airway management during anesthesia [98,103,104], although some studies report ongoing difficulties with intubation despite ERT and HSCT [105] especially in older patients [106]. These difficulties may be due to continued musculoskeletal issues and MPS-related pathology. Beneficial effects of HSCT exceed those of ERT, postulated to be the result of developing inhibitory IDUA antibodies, as patients without high levels of such antibodies showed significant clinical improvement [98]. However, ERT reduced the frequency of upper airway infections and improved sleep apnea [51,52,107,108,109]. Results from a case study demonstrated that relapse in respiratory function could be treated by weekly ERT with good clinical outcome [110].

Few studies address the effect of treatment on pulmonary function. A large retrospective study of patients treated with ERT and/or HSCT showed improvement or stabilization of pulmonary function in the majority of patients for the duration of over 12 years. However, residual restrictive lung disease remained in all patients and 1/3 of the patients experienced progressive loss of pulmonary function despite treatment [111]. Other studies reported ongoing overnight hypoxia, suggesting incomplete resolution of pulmonary insufficiency [29]. Patients with MPS I are at an increased risk of developing pulmonary complications following HSCT. ERT alone resulted in improved lung function [51,107,112]; however, this appeared to be only a temporary effect [51]. Part of the improved pulmonary function was thought to result from reduction in liver and/or spleen size after HSCT and ERT, as this would reduce pressure on the diaphragm. It remains to be seen whether the variety of responses were caused by donor/recipient specific factors or by the persistent disease-associated skeletal deformities of the thoracic cage, which are not affected by either treatment. Combined HSCT and ERT therapy had beneficial outcomes on pulmonary manifestations [76].

To date few studies have evaluated the effect of treatment on respiratory issues in animal models of MPS I [113], possibly due to the often challenging assays involved.

### 3.3. Hearing Loss

Hearing loss is one of the most common manifestations in patients with MPS I. Hearing loss can be neurosensorial, conductive, or present as a mixture of both. Some improvement or stabilization of hearing loss was observed for the majority of children treated by HSCT, especially in children receiving the transplant at an early age [11,85,114,115,116]. This improvement was attributed predominantly to improved sensorineural hearing [85,115]. A longitudinal follow-up study of 28 patients with MPS I post-HSCT investigated hearing loss using auditory brainstem response and pure tone audiometry [117]. While some improvement was noted, none of the patients recovered normal hearing. Importantly, the authors discovered that initially improved hearing was due to improved air conduction thresholds, possibly by a reduction in GAG deposits in the middle ear, and reduced frequency of chronic otitis. However, bone conduction worsened over time and resulted in an overall loss of hearing, which became apparent 10 years after transplantation. These results confirmed the finding of Aldenhoven reporting overall hearing loss in a large MPS I cohort 10 years post HSCT [29].

The effect of ERT on hearing loss is unclear. Case studies reported stabilized sensorineural hearing loss in one patient and improved conductive hearing loss in the other [118], or progressive sensorineural hearing loss [119]. A larger longitudinal study of 15 patients with MPS I treated with ERT reported continuous hearing loss over 2.5 years [120].

### 3.4. Skeletal Manifestations

Skeletal manifestations in MPS I are notoriously resistant to treatment, necessitating patients with MPS I to undergo surgeries including correction of genua valga, odontoid hypoplasia, hip dysplasia, and thoracolumbar kyphosis. As discussed in other parts of this review, remaining bone deformities may cause other complications such as limiting pulmonary function due to severe kyphosis and/or scoliosis.

The first report investigating musculoskeletal manifestations after HSCT showed promising results [35] and following studies indicated normalized longitudinal growth for the first year after HSCT [121]. Eventually, however, long-term follow-up studies revealed that the initial normalization of the growth rate was followed by a subsequent growth failure [11,29,34,85,122,123,124]. For example, while the longitudinal growth in children with MPS I receiving HSCT was better as compared to non-treated historic controls, it remained significantly lower when compared to CDC growth charts [125]. The reason for eventual decline in growth rate is unknown. It has been posited to be due to a reduced ability of enzyme to penetrate the epiphyseal growth plate as chondrocytes differentiate and become ossified [122]. Reduced trunk growth with continuous kyphosis can contribute to short stature [85,122,123,126], although poor growth was described irrespective of kyphosis [11].

Aside from impaired longitudinal growth, hip dysplasia [61,63,127,128], genu valgum [63,129], thoracolumbar kyphosis, and scoliosis [61,63,129,130,131] progress despite HSCT. Human ex vivo data is limited to a single publication [132] where two patients were briefly described, and histologic evidence was provided for correction of chondrocytes from the lumbar spine in a patient with MPS I treated with HSCT. The lack of effectiveness on skeletal manifestations through ERT and HSCT has been attributed to several factors. It is possible that the amount of enzyme that reaches ossification centers may simply be too low to overcome the accumulation of GAG and associated defects [86,122]. Moreover, cultured mouse osteoblasts, derived from a mouse calvaria immortalized precursor cell line, have decreased uptake of exogenous IDUA compared to mouse fibroblasts [133].

Remarkably, disease manifestations in facial bones [85,122] and the odontoid process showed improvement following HSCT. The typical coarse facial features were alleviated, the head circumference normalized [11,29], and prevalence of odontoid hypoplasia was reduced [63,126,129,134,135,136], with much desired relief of spinal cord compression. It is unclear why these bones responded to HSCT treatment compared to the bony skeleton. However, the distinct difference between facial and cranial bones versus long bones and vertebrae is that they form through intramembranous ossification versus endochondral ossification.

ERT alone has minimal effects on skeletal manifestations, especially in patients with Hurler syndrome [112,137]. Better outcomes were reported in patients with attenuated MPS I, where early initiation of ERT treatment prevented or delayed development of skeletal manifestations [138,139].

Combination of HSCT and ERT may provide the best outcome. In a recent clinical trial the effect of ERT post-HSCT was evaluated and resulted in improved growth rate, particularly in young patients [67]. Peri-HSCT ERT resulted in improved odontoid process morphology and reduced spinal cord compression in a 10-year old girl treated with HSCT and ERT at age 18 months [74].

In animal models of MPS I, treatment with HSCT showed overall more promising outcomes, especially when performed in neonatal animals. Less severe skeletal manifestations were observed in MPS I cats and dogs post-HSCT [40,90]. Transplantation of neonatal MPS I mice with umbilical cord blood or BM prevented the development of dysostosis multiplex altogether [41,42]. Notably, these studies corrected for busulfan-associated effects by including busulfan-treated wildtype mice that received wildtype HSCT. Dimensions of skull and leg bones, and cortical and trabecular bone architecture of treated animals were similar to that of wild-type control mice. On the cellular level, osteocytes showed fewer and smaller GAG deposits, and the growth plates appeared more organized as compared to untreated animals [41]. Transplantation at a later age (8 weeks) resulted in overcorrection of the reduced osteoclastogenesis described in untreated IDUA-deficient mice, which was corrected with combined HSCT and ERT [140]. ERT in neonatal MPS I mice was not accompanied by any improved bone manifestations compared to non-treated animals [141], while in MPS I dogs ERT attenuated skeletal manifestations, especially when administered to neonatal animals [136]. Particular benefit was reported regarding cervical spine disease.

### 3.5. Joint Mobility

Restriction of joint range of motion (JROM) of upper and lower limbs is present in many patients with MPS I and restricts their mobility [142,143,144]. Some studies reported that HSCT had a beneficial effect on JROM. It remains to be determined whether all joints benefit from HSCT [11,29,63,122,145]. Specifically, the effect on shoulder mobility varied between different studies. Notably, range of motion of the shoulder joint responded positively to ERT, while other joints showed little effect [51,54,57,146]. The effect of HSCT on development of carpal tunnel syndrome remains unclear, with some studies indicating partial improvement after HSCT, especially if performed at a young age [147], while others found no effect of HSCT with or without ERT on the development of carpal tunnel syndrome [148]. Studies in mice found no benefit of ERT on joint disease [141].

### 3.6. Cardiac Function

Cardiac manifestations in patients with MPS I involve mainly valvular heart disease and coronary artery narrowing. However, occlusion of the abdominal aorta and renal arteries and associated systemic hypertension have also been described. Animal models showed aortic dilation as well.

HSCT improved some of the underlying cardiac manifestations, including cardiac hypertrophy [15,149,150,151] and coronary artery narrowing [10,152]. However, mitral and aortic valve insufficiencies persisted [7,15,153], causing progressive valvular dysfunction, including stenosis and regurgitation [7,15]. Similar to changes in skeletal manifestations and corneal clouding, insufficient correction of valvular disease was likely due to inadequate supply of the heart with enzyme in patients. In contrast to these findings in patients, BMT in MPS I dogs resulted in partial correction of mitral and aortic valve abnormalities and correction of aortic root dilation [90,154]. The differences in response are not understood.

ERT had a positive effect on cardiac hypertrophy and stabilized or even improved systolic ventricular function [53,155,156,157]. However, ERT did not prevent progression of cardiac valve thickening [51,155,156,158,159,160], although some studies reported stabilization or improvement of cardiac valve function [51,107,141,161]. Similar results were reported in MPS I dogs receiving ERT after a tolerization regimen comprising cyclosporine A with azathioprine and low-dose IDUA [162].

### 3.7. Cognitive Function

HSCT remains the standard treatment for severe MPS I because it can stabilize cognitive function and prevent progressive developmental decline [16,83,163,164,165]. However, HSCT cannot restore cognitive function. Consequently, neurodevelopmental outcome is predicted by baseline function and age at HSCT. Typically, cognitive function declines in the first year post-HSCT, after which it stabilizes [8,50,153], possibly due to a delay in infiltration and engraftment of donor cells in the CNS. The relative long lag-time between treatment and possible effect on cognitive function poses a challenge for clinical trials, which is also augmented by the interpatient variability of cognitive manifestations.

ERT on its own is not expected to provide neurocognitive benefit due to the inability of the enzyme to cross the BBB. However, there is a single case report of a girl with Hurler syndrome who was treated with only ERT starting before age 2, showing a significantly extended course of normal neurocognitive function, unpredicted by the natural history of the disease; the authors speculated that ERT was a critical factor in this finding [50,66]. Further, in MPS I mice, treatment with doses of ERT at levels higher than the clinical standard dose was accompanied by an increase of IDUA activity and reduction in GAG accumulation in the brain [141,166,167]. It is interesting that the treatment not only prevented the development of cognitive impairment in neonatal animals, but also resulted in a significant improvement when administered up to 10 weeks of age. However, older animals with established cognitive impairment, at 6 months or older, did not show improved neurobehavior [161].

The observation that high-dose ERT resulted in an increase of IDUA activity in the brain may indicate that the BBB is not impermeable to enzyme, but that the amount of enzyme delivered under clinical dosages is insufficient to yield detectable changes. Combined HSCT and ERT was associated with better short-term cognitive function in patients with MPS I [50]. The mechanism remains unclear; however, it has been posed that conditioning may increase BBB permeability [168], thereby allowing entry of IDUA into the CNS.

## 4. Experimental Therapies

The residual disease burden in patients with MPS I is caused mainly by insufficient enzyme levels in the CNS, the eyes, heart, and bones. Different experimental therapies have been developed to address this shortcoming, and some preclinical treatments have been translated into clinical trials (Table 2).

### 4.1. Substrate Reduction

While the majority of current therapies focus on the delivery of IDUA, a different approach is taken in substrate deprivation therapy. Treatment of 4 week old MPS I mice with weekly IV injections of the GAG synthesis inhibitor Rhodamine B resulted in prevention of some skeletal manifestations and better cognitive function as evaluated by a water cross maze [169].

### 4.2. Accelerated GAG Degradation

A different approach was taken by stimulating autophagy in order to accelerate GAG degradation [170]. Resveratrol is a stilbenoid polyphenol with many biological properties, including acceleration of autophagy, which may reduce GAG accumulation in patients with MPS I. Importantly, Resveratrol crosses the blood brain barrier. However, the short half-life of the drug and its rapid degradation in the liver limits its applicability as a long-term treatment. Jupiter Orphan Therapeutics reported that administration of their formulation of Resveratrol (JOTROL™) to rats led to an increase of IDUA levels in plasma and brain [171].

### 4.3. Anti-Inflammatory Therapy

Subcutaneous injections of the anti-inflammatory drug pentosan polysulfate (PPS) in MPS I dogs not only reduced inflammatory markers, but resulted in reduced GAG concentrations, increased luminal openings, and reduced intimal media thickening in the carotid arteries and aortas of MPS I dogs [172]. In a subsequent study of patients with attenuated MPS I, this treatment resulted in reduced urinary GAG excretion, improved joint mobility, and reduced pain [173]. Treatment of one patient with MPS I with the anti-inflammatory adalimumab resulted in improved range-of-motion and reduced bodily pain (NCT02437253) [174]. A phase 1/2 trial (NCT03153319) is currently recruiting patients with MPS I to evaluate safety and efficacy of adalimumab.

### 4.4. Intracerebroventricular and Intrathecal Delivery ERT

Direct delivery of enzyme by intrathecal (IT) ERT in dogs was associated with partial correction of pathological manifestations as assessed by neuroimaging [175]. Similar to direct enzyme delivery, transplantation of HSCT into the cerebral ventricle of neonatal immunodeficient MPS I mice resulted in improved motor function as assessed by rotarod test [176].

A number of small clinical trials are currently underway to test IT ERT on cognitive function in patients with MPS I. The first intrathecal ERT in patients with MPS I resulted in some improvement in endurance and pulmonary function [177]. IT ERT in one patient with attenuated MPS I resulted in significantly improved memory and adaptive functioning and mildly improved attention and IQ [66]. IT IDUA administration in five patients with MPS I (NCT00215527, NCT00786968) resulted in subjective improvement reported in the three subjects that completed the study. However, objective outcome measures including the 6 min-walk-test did not support this finding [178]. Similar findings were seen in a follow-up study of 16 patients with MPS I (NCT00852358) [179]; however, the authors argued that the lack of effect may have been due to the relatively older age, normal cognitive function of the participants, and the short study period of 2 years. More promising results were reported when children with MPS I were infused intrathecally with Laronidase prior to and following HSCT (NCT00638547). Here, improvement was also reflected in decreased CSF opening pressure and levels of biomarkers of disease activity and inflammation. Notably, some of these changes occurred after ERT-IT, but prior to HSCT. Further, this study reported a significant relationship between biomarker change and neurocognitive outcome, in that a reduction in a heparan sulfate-derived part of the GAG was associated with a more favorable IQ trajectory [180].

### 4.5. In Utero ERT Treatment

Treatment in utero was attempted in the canine MPS I disease model, where fetal pups were injected either with retroviral vector containing the IDUA encoding sequence, or with HSC retrovirally transduced with IDUA. While transduction and cell engraftment were observed, no IDUA activity or IDUA transcripts were detected; consequently there was no evidence of disease amelioration [181,182]. A clinical phase I study is planned to establish safety and feasibility of in utero ERT to induce tolerance and improved neurodevelopmental outcomes (NCT04532047).

### 4.6. Shuttling of Enzyme Across the BBB

Tissue-specific delivery of IDUA to the brain was attempted by targeting receptors present on the luminal side of the BBB. Fusion of the IDUA molecule to ApoE-derived receptor-binding peptides enabled the enzyme to cross the BBB of MPS I mice resulting in increased brain IDUA activity and normalized brain GAG levels [183]. The treatment correlated with normalization of behavioral performance as assessed by repeated open-field tests [167]. The same receptor pathway was targeted by an Angiopep-2-IDUA fusion protein, developed by Angiochem Inc. An unusual delivery pathway was selected by fusing IDUA to the plant lectin ricin B chain (RTB). Peripheral delivery of RBT-IDUA led to a significant increase in enzyme activity and normalization of GAG levels in the brain and improved cognitive function in MPS I mice [184]. BioStrategies LC is developing the RTB-IDUA drug as a delivery option to treat bone and connective tissue in MPS I. Finally, fusion of IDUA to a monoclonal antibody targeting the human insulin receptor [185] allowed transport of IDUA across the BBB in animal models [186,187]. The effect of this approach on cognitive function was investigated in a 52-week clinical trial enrolling children with MPS I with severe neurocognitive impairment (NCT03071341). An additional trial (NCT03053089) enrolled patients with attenuated MPS I. Treatment stabilized CNS function as assessed by cognitive testing and total grey matter volume [188].

Another IDUA fusion protein targeted to cross the BBB was developed by JCR Pharmaceuticals Co Ltd. (JR 171). A phase 1/2 clinical trial (NCT04227600) was concluded in July 2020; however, results are unpublished. An additional phase 1/2 trial NCT04453085 is planned, but not yet recruiting. While the exact nature of the modification is proprietary, it is likely that the fusion protein consists of IDUA fused to anti-human transferrin receptor analogous to the company’s drug JR-141, a fusion protein of anti-human transferrin receptor and iduronate-2-sulfate [189]. The human transferrin receptor was also targeted by the transferrin-IDUA fusion protein [190], developed as Txb4-Ls1 by Ossianix.

### 4.7. Molecular Therapies

#### 4.7.1. Nonsense Suppression and mRNA Engineering

The most common mutation in MPS I is the W402X nonsense mutation, which introduces a premature stop codon. Nonsense suppression therapy aims at the suppression of the nonsense mutation, thereby enabling continuous translation into a full-length protein. Suppression can be achieved using specific drugs [191]. The aminoglycoside derivative NB84 partially restored IDUA activity in MPS I mice carrying the murine nonsense mutation (W392X mice) and resulted in partial prevention of cardiac, skeletal, and behavioral abnormalities [192]. The effect on behavioral manifestations was likely due to aminoglycosides entering the CNS at ~10–20% of their serum concentration.

Therapeutic RNA editing aims to correct a mutation within the mRNA. In one approach, a synthetic antisense RNA complementary to the mutated sequence forms a double-stranded RNA, which activates the deamination of adenosine-to-inosine (A-to-I), resulting in a read-through of the stop-codon. The deamination is carried out by endogenous adenosine deaminase acting on RNA (ADAR) enzymes, which act on double stranded RNAs.

To date, results for therapeutic RNA editing to correct the TGG>TAG nonsense mutation within the IDUA gene have been published only in abstract format and will be discussed briefly here. The Leveraging Endogenous ADAR for Programmable Editing of RNA (LEAPER) program uses ADAR recruiting (ar) RNA, which targets the enzyme to the TGG>TAG nonsense mutation. Administration of arRNA packaged into AAV vectors in MPS I mice carrying the W392X mutation resulted in higher IDUA enzymatic activity [193]. This approach is currently under investigation for its applicability as an MPS I therapeutic by EdiGene Inc. A similar approach is used by RNA Editing for Specific C-to-U Exchange (RESCUE), which fuses the inactive Cas13 (dCas13) protein to the ADAR deaminase domain (ADARdd). Delivery of this fusion protein and guide RNA via AAV into W392X mice resulted in correction of 15% of total IDUA mRNA and a significant increase of IDUA activity in the liver [194]. In a slightly different strategy, exogenous adenosine deaminases were introduced into the host. Fusion of two monomers of *E.coli* tRNA-specific adenosine deaminase with inactive Cas9 protein gives rise to the Adenine Base Editor (ABEmax). AAV9 mediated co-delivery of guide RNA and ABEmax into neonatal W392X mice resulted in durable enzyme expression alongside reduction of accumulation of GAGs in tissues [195].

Systemic delivery of suppressor tRNA therapy by AAV vectors into W392X mice resulted in prolonged restoration of low serum and liver IDUA activity (1–3% of the normal level) [196].

#### 4.7.2. Ex Vivo Gene Transfer

Stable genetic complementation of IDUA deficiency in autologous HSC has the potential to address some of the complications of allotransplant, such as graft-vs-host disease, while at the same time engineering a much higher level of enzyme expression and delivery and maintaining access to the CNS [197,198]. Transplantation of HSPCs transduced ex vivo with IDUA-lentivirus vector into 8 week old MPS I mice resulted in the correction of disease manifestations, including cognitive impairment, hearing deficits, skeletal deformities, and retinopathy [197]. Results from studies investigating this approach in MPS I dogs using murine gamma-retroviral vector were less compelling [199,200]. While the engineered cells showed high IDUA expression and transplanted cells engrafted successfully in the animals, enzyme expression in vivo was undetectable. This gene deactivation was caused by cellular and humoral immune responses against the transduced cells and the resultant protein.

The ex vivo gene therapy approach is currently being tested in a small clinical trial, where patients with severe MPS I received HSPCs transduced with lentiviral vector encoding IDUA (NCT03488394). Preliminary data after 1 year suggested stabilization of cognitive function and improvement of skeletal deficits [201].

Transplantation of 6–8 week old immune-deficient MPS I mice with HSCPs edited via the CRISPR-Cas9 approach for insertion of the IDUA coding sequence into the CCR5 locus resulted in IDUA activity in serum, liver, spleen, and brain together with normalization of skeletal parameters and improved cognitive function [202].

The Sleeping beauty (SB) transposon system presents another alternative to viral genetic therapy for stable gene transfer and expression. The SB system uses the SB transposase to integrate gene sequences in co-delivered transposons into host chromosomal DNA. Hydrodynamic co-delivery of SB transposase and the SB transposon encoding IDUA into MPS I mice or MPS I dogs resulted in successful expression of the enzyme in the liver. However, the animals’ immune response resulted in the subsequent decline of IDUA levels [203,204]. Immunosuppression or the use of immunodeficient NOD/SCID-MPS I mice allowed prolonged expression of IDUA in the liver, reduction of GAG accumulation, and correction of some skeletal manifestations [205]. Immusoft Corp is currently developing the SB approach as a cell-based therapy for IDUA deficiency using autologous human B cells. Infusions of immunodeficient NSG-MPS I mice with IDUA-transposed B cells resulted in substantial plasma IDUA activity and reduction of tissue GAG storage [206]. A similar approach was taken using a combination of CRISPR/Cas9 and rAAV vector to integrate the IDUA gene into B cells. Transplantation of these IDUA-positive B cells into NSG-MPS I mice resulted in a significant increase in IDUA enzyme activity [207].

Sigilon Therapeutics Inc is currently developing their Shielded Living Therapeutics approach as a strategy to protect therapeutic human cells that have been engineered to express high levels of IDUA from the host’s immune response. Administration of this shielded IDUA-secreting human cell line (SIG-005) into MPS I mice resulted in GAG reduction in plasma and tissues [208].

#### 4.7.3. In Vivo Gene Transfer

In vivo gene therapy using AAV vector can take advantage of varying tropisms conferred by different AAV serotypes to direct transduction to a desired tissue. However, AAV rarely integrates into the host genome, so these vectors are most applicable to target tissues where there is limited cell division, such as the CNS. Intrastromal delivery of AAV8G9-vector encoding IDUA resulted in lasting reversal of corneal clouding in MPS I dogs [209]. Administration of AAV2 vector encoding IDUA in neonatal mice partially ameliorated bone defects and behavioral abnormalities as assessed in an open field test [210]. There was only a moderate level of IDUA detected in brain tissues, so these findings suggest that even small amounts of enzyme in the CNS can have a significant impact on cognitive function. Much higher levels of enzyme were expressed in the brain after ICV infusion of neonatal MPS I mice with AA8 vector encoding human IDUA, preventing emergence of neurocognitive deficits as determined in the Morris water maze [211]. Intranasal administration of IDUA-encoding AAV9 vector to adult MPS I mice resulted in normalized IDUA enzyme activity and reduction of GAG accumulation in the brain. Moreover, neurocognitive function was corrected as assessed in the Barnes maze [212]. AAV9 vector encoding IDUA corrected CNS pathology after intrathecal delivery in MPS I cats [213]. IDUA transduction using AAV9 vector was tested for safety and expression in non-human primates after intrathecal suboccipital delivery to the CSV [214]. REGENXBIO is currently testing intracisternally administered rAAV9 vector encoding IDUA in a clinical trial for MPS I (NCT03580083).

Administration of gamma-retroviral vector encoding IDUA into neonatal MPS I mice prevented development of ocular and hearing impairments, aortic insufficiencies, and skeletal defects [215] and resulted in improved outcomes in adult mice as well [216].

In vivo gene therapy using the CRISPR-Cas9 strategy was first tested in neonatal MPS I mice, where the ubiquitously expressed ROSA26 locus was targeted as the transgene insertion site. A long-term, modest increase in IDUA activity in serum and all organs (except for brain) was observed using this approach [113]. Pulmonary function was normalized, bone defects were prevented, and elastin breaks in the aorta were partially reduced. However, heart valves and brain showed no functional improvement [113]. A Zinc Finger Nuclease (ZFN) mediated gene editing approach was used to insert the IDUA coding sequence into the albumin locus for high-level protein expression [217]. Delivery of the construct by AAV8 vector to hepatocytes in 4–10 week old MPS I mice resulted in significant increases in IDUA activity and GAG reduction in blood and peripheral tissues and in the brain. Development of neurobehavioral deficits was prevented as assessed by Barnes maze. An ongoing phase 1/2 clinical trial involving patients with mild forms of MPS I with no CNS involvement is being sponsored by Sangamo Therapeutics (NCT02702115).

## 5. Conclusions

The current approved treatment options for MPS I consist of HSCT, ERT, or combinations of both therapies. While these treatments significantly improve disease manifestations and prolong life, a considerable burden of disease remains in the treated children and adults. Both treatments may at best prevent the development or worsening of abnormal function and somatic complications but cannot revert already existing symptoms. Therefore, treatment must commence as early as possible for maximum effect. Even when conducted under optimal conditions, such as early age and matching non-carrier donor for the HSCT, the therapeutic effect in some systems appears to wear off after several years. Some organs are altogether resistant to treatment. The most resistant organs/tissues are the bones, eyes, and heart valves.

A number of experimental strategies are currently under development to reduce the burden of disease further. Some molecular therapies target nonsense mutations at the mRNA level through nonsense suppression, mRNA editing, and suppressor tRNAs. Gene transfer is being tested both ex vivo by engineering HSCPs and human B cells prior to transplantation and in vivo using both viral and non-viral approaches. Other therapies address GAG accumulation by enhancing autophagy or inhibiting GAG synthesis. The effect of delivering IDUA to the CNS via direct infusion, shuttling pathways, or targeted viral delivery is currently under investigation in small clinical trials. With concentrated efforts, the prospect for significantly improving the long-term outcome for these children is within our grasp.

## Figures and Tables

**Table 1 biomolecules-11-00189-t001:** Effect of HSCT and ERT on clinical manifestations in MPS I.

Clinical Manifestation	HSCT	ERT	HSCT + ERT
**Partial improvement with added benefit of combination therapy**			
Cognitive function	Stabilization	No effect	Improvement
Pulmonary function	Limited improvement	Improvement	Improvement
Skeletal manifestations	Minimal effect of linear growth Improved facial features and odontoid hypoplasia	No effect	Improved growth rate
**Partial improvement**			
Upper respiratory	Improvement	Improvement	Improvement
Joint mobility	Improved range of motion	Improved range of motion (shoulder)	NA
Cardiac function	Improved cardiac hypertrophy and coronary artery narrowing No effect on valve insufficiencies	Improved cardiac hypertrophy and ventricular function No significant effect on cardiac valve thickening	Improvement
**Limited effect**			
Hearing loss	Improvement/ stabilization	No effect	NA
Corneal clouding	Limited stabilization	No effect	NA
Retinal dysfunction	No effect	No effect	NA
Hearing loss	Improvement/ stabilization	No effect	NA

**Table 2 biomolecules-11-00189-t002:** Clinical Trials for MPS I.

Drug	Clinical Trials
**Anti-inflammatory therapy**	
Adalimumab	Phase I/II: NCT02437253: completed NCT03153319: recruiting
**In utero ERT**	
laronidase	Phase I: NCT04532047: not yet recruiting
**Intrathecal delivery**	
laronidase	Phase I: NCT00215527, NCT00786968: terminated due to slow enrolment) Phase not applicable: NCT00852358: completed NCT02232477: terminated due to COVID-19
laronidase with HSCT	Phase I: NCT00638547: completed
**BBB-crossing IDUA-fusion proteins**	
AGT-181 (fusion to Insulin receptor monoclonal antibody)	Phase I/II: NCT03071341, NCT03053089, NCT02597114: completed
JR-171 (Undisclosed fusion partner)	Phase I/II: NCT04227600, NCT04453085: not yet recruiting
**Ex vivo gene transfer**	
Autologous HSPC transduced with IDUA	Phase I/II: NCT03488394: recruiting
ISP-001 (B cells transposed with IDUA)	Phase I/II: NCT04284254: not yet recruiting
**In vivo gene transfer**	
RGX-111 (AAV9-mediated)	Phase I/II: NCT03580083: recruiting
SB-318 (Genome editing ZFN)	Phase I/II: NCT02702115: active, not recruiting

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
