# Peer review of "Mucopolysaccharidosis Type I: Current Treatments, Limitations, and Prospects for Improvement"

_biomolecules, 2021, doi:10.3390/biom11020189_

Round 1

Reviewer 1 Report

The review by Hampe et al is a comprehensive review of available therapies and their impact on clinical outcomes. It is well referenced and refers to the current active/discontinued studies.

Some comments: it would be worth describing the long-term impact of chemotherapy on hormonal dysfunction in MPS I patients who underwent HSCT 20-30 years ago and were exposed to the large doses on chemo-and radio-therapy. The long-term impact is important as it affects their fertility. There is some data published in the last few months/years and worth referencing it.

reference 30- small typo needs to be corrected

Page 5, line 215- MPS, not MSP, please correct.

Otherwise, a concise paper with several new aspects regarding the mechanisms of action of new therapies.  

Author Response

  • it would be worth describing the long-term impact of chemotherapy on hormonal dysfunction in MPS I patients who underwent HSCT 20-30 years ago and were exposed to the large doses on chemo-and radio-therapy. The long-term impact is important as it affects their fertility. There is some data published in the last few months/years and worth referencing it.

We agree that the long-term effect of myeloablative conditioning on fertility needs to be included and have added the following section: “Long-term effects of myeloablative conditioning regimens are infertility in both females and males [18,19]. Comparing infertility risks associated with treatment in pre-puberty (1-12 years) with treatment after the age of 13, revealed opposite trends in males and females. While pre-pubertal treatment in males was associated with increased risk for infertility, females treated at a younger age had a lower risk for infertility [20]. Busulfan-based treatment was associated with higher infertility in females but not in males, and TBI increased risk of infertility in males, but not in females [20]. Fertility preservation needs to be addressed in patients prior to HSCT [21].

  • reference 30- small typo needs to be corrected

This has been corrected.

  • Page 5, line 215- MPS, not MSP, please correct.

Reviewer 2 Report

This is a nice review by a very competent group. It gives a good overview of the theme, and deserves publication. My only suggestion would be to include either another table or a figure in the manuscript, that summarizes approaches, pros-, cons, limitations of treatments. 

Some minor corrections:

"It is important to note that there is a great deal of variability in the ocular pathology in individuals with MSP I" ---- I believe it is MPS I

"To date few studies have evaluated the effect of treatment on respiratory issues in animal models of MPS I"----- Schuh et al (gene therapy, 2020) have shown that MPS I mice have respiratory issues and that gene editing improves this parameter. This reference is even in the manuscript already, but could be added here maybe.

Author Response

My only suggestion would be to include either another table or a figure in the manuscript, that summarizes approaches, pros-, cons, limitations of treatments. 

We included an additional table summarizing the impact of HSCT, ERT and their combined use:

Clinical manifestation

HSCT

ERT

HSCT+ERT

Partial improvement with added benefit of combination therapy

Cognitive function

Stabilization

No effect

Improvement

Pulmonary function

Limited improvement

Improvement

Improvement

Skeletal manifestations

·  Minimal effect of linear growth.

·  Improved facial features and odontoid hypoplasia

No effect

Improved growth rate

Partial improvement

Upper respiratory

Improvement

Improvement

Improvement

Joint mobility

Improved range of motion

Improved range of motion ( shoulder)

NA

Cardiac function

·  Improved cardiac hypertrophy and coronary artery narrowing.

·  No effect on valve insufficiencies

·   Improved cardiac hypertrophy and ventricular function.

·   No significant effect on cardiac valve thickening.

Improvement

Limited effect

Hearing loss

Improvement/ stabilization

No effect

NA

Corneal clouding

Limited stabilization

No effect

NA

Retinal dysfunction

No effect

No effect

NA

Hearing loss

Improvement/ stabilization

No effect

NA

Some minor corrections:

  • "It is important to note that there is a great deal of variability in the ocular pathology in individuals with MSP I" ---- I believe it is MPS I

This has been corrected.

  • "To date few studies have evaluated the effect of treatment on respiratory issues in animal models of MPS I"----- Schuh et al (gene therapy, 2020) have shown that MPS I mice have respiratory issues and that gene editing improves this parameter. This reference is even in the manuscript already, but could be added here maybe.

 This reference has been added.

Reviewer 3 Report

line 24: edit "...and attenuated (Hurler-Scheie, Scheie) forms. (list in order of severity)

line 29:  Here you are basically claiming that treatment can prevent clinical manifestations. The remainder of your manuscript does not reflect this, so the beginning of your paper is misleading and confusing.  HSCT definitely has been reported to improve the outcome/reducing the progression, but not preventing all of the symptoms of MPS I. Reword to focus on prevention of neurologic deterioration. 

line 51: change Hurler 'patients' to Hurler syndrome (suggestion for patient first language)

line 54:  change 'improved facial features' to 'reduced facial coarseness'

line 62/63 - I do not think you can say this.  see Parini et al (2017) - https://www.ncbi.nlm.nih.gov/pmc/articles/PMC5472858/

section 2.1.1 - I think it would be appropriate here to at least mention pre-HSCT  ERT conditioning to improve clinical conditions pre-procedure.  Then you can expand as done in lines 165-172.

lines 95/96 - I think you can combine 'IDUA activity levels and IDUA-enzyme activity in lymphocyte lysates' (is [23] an appropriate reference here?)

line 325 - capitalize IDUA (not sure why it is italicized?)

section 4.2 - description of Resveratrol should be given.  Seems confusing the way this paragraph reads.

line 603/4 - ' ...under development to further reduce disease burden'

General comments:

Much needed article addressing all of the treatment modalities currently in use or being studied in MPS I.  I think this will be a very helpful reference paper for those caring for patients.  I'd recommend putting a section of combined HSCT and ERT between the two separate sections, as this is a very common practice and should be addressed.  I also think you should mention something about diagnostic delay and how this contributes to disease burden and subsequent treatment response, then mentioning the solution to this being newborn screening.  I would also really like to see you change the language throughout the manuscript to reflect 'people first' vs 'disease first' (patients with MPS I vs MPS I patients).

Author Response

  • line 24: edit "...and attenuated (Hurler-Scheie, Scheie) forms. (list in order of severity)

This has been corrected.

  • line 29:  Here you are basically claiming that treatment can prevent clinical manifestations. The remainder of your manuscript does not reflect this, so the beginning of your paper is misleading and confusing.  HSCT definitely has been reported to improve the outcome/reducing the progression, but not preventing all of the symptoms of MPS I. Reword to focus on prevention of neurologic deterioration. 

We agree that HSCT does not fully prevent clinical manifestations and have amended the language accordingly.

  • line 51: change Hurler 'patients' to Hurler syndrome (suggestion for patient first language)

We appreciate raising this concern. The terms “Hurler patients” and “MPS I patients” have been changed to “patients with Hurler syndrome” and “patients with MPS I” throughout the manuscript.

  • line 54:  change 'improved facial features' to 'reduced facial coarseness'

This has been changed.

  • line 62/63 - I do not think you can say this.  see Parini et al (2017) - https://www.ncbi.nlm.nih.gov/pmc/articles/PMC5472858/’

We modified the sentence as follows: “Tissue specific improvement is caused by different accessibility of tissues, such as heart, eyes, and bones, to circulating enzyme.”

  • section 2.1.1 - I think it would be appropriate here to at least mention pre-HSCT  ERT conditioning to improve clinical conditions pre-procedure.  Then you can expand as done in lines 165-172.

We added the following paragraph after ERT treatment:

“While ERT is not recommended as the sole treatment for Hurler syndrome, a combination of ERT with HSCT may have benefits over each treatment alone [71–73]. Peri-transplant ERT appears to have beneficial effects on the clinical condition of the patient [46,56,74]. ERT can bridge the time until a suitable donor for HSCT has been identified and therefore ERT is often initiated at time of diagnosis [28]. Importantly, ERT was not associated with a reduced engraftment rate [46,71,74,75], and subsequent HSCT attenuated the formation of neutralizing IDUA antibodies [76]. Moreover, GAG-reduction due to peri-transplant ERT appeared to improve HSCT engraftment [71,73,77]. Continuous ERT post-transplant has been reported to improve residual disease burden [48,56,65,72,74]. Recent studies in neonatal MPS I mice allowed an in-depth analysis of the combined treatement [78]. Animals receiving both HSCT and ERT showed higher IDUA levels in the spleen, lower plasma GAG levels, and improved bone architecture compared to animals receiving either treatment alone [78]. Beneficial effects of combined HSCT/ERT treatment pertaining to specific manifestations are discussed in the appropriate sections below. “

  • lines 95/96 - I think you can combine 'IDUA activity levels and IDUA-enzyme activity in lymphocyte lysates' (is [23] an appropriate reference here?)

This has been changed. The reference has been replaced with a more appropriate paper.

  • line 325 - capitalize IDUA (not sure why it is italicized?)

This has been corrected.

  • section 4.2 - description of Resveratrol should be given.  Seems confusing the way this paragraph reads.

We added a description of the biological function of Resveratrol regarding MPS I treatment.

  • line 603/4 - ' ...under development to further reduce disease burden'

This has been corrected.

General comments:

Much needed article addressing all of the treatment modalities currently in use or being studied in MPS I.  I think this will be a very helpful reference paper for those caring for patients. 

  • I'd recommend putting a section of combined HSCT and ERT between the two separate sections, as this is a very common practice and should be addressed. 

See our response to point 6.

  • I also think you should mention something about diagnostic delay and how this contributes to disease burden and subsequent treatment response, then mentioning the solution to this being newborn screening. 

We added the following text to the introduction: “Therefore, treatment must commence as early as possible for maximum effect and diagnostic delay – due to the nonspecific nature of early symptoms -  limits treatment success [1]. To overcome this limitation, implementation of MPS I in newborn screening programs is strongly recommended [2,3].

  • I would also really like to see you change the language throughout the manuscript to reflect 'people first' vs 'disease first' (patients with MPS I vs MPS I patients).

See our response to point 3.